# Structure Simulation and Equilibrium Evaluation Analysis of Regional Water Resources, Society, Economy and Ecological Environment Complex System

**DOI:** 10.3390/e25020181

**Published:** 2023-01-17

**Authors:** Chengguo Wu, Xiaoyu Wang, Juliang Jin, Yuliang Zhou, Xia Bai, Liangguang Zhou, Fang Tong, Libing Zhang, Yi Cui

**Affiliations:** 1School of Civil Engineering, Hefei University of Technology, Hefei 230009, China; 2Institute of Water Resources and Environmental Systems Engineering, Hefei University of Technology, Hefei 230009, China

**Keywords:** water resources spatial equilibrium system, connection number, ordered degree entropy, system dynamics, Hefei city

## Abstract

Currently, the implementation of water resource spatial equilibrium strategy is a fundamental policy of water resource integrated management in China; it is also a considerable challenge to explore the relationship structure features of water resources, society, economy and ecological environment (WSEE) complex system. For this purpose, firstly, we applied information entropy, ordered degree and connection number coupling method to reveal the membership characteristics between different evaluation indicators and grade criterion. Secondly, the system dynamics approach was introduced to describe the relationship features among different equilibrium subsystems. Finally, the ordered degree, connection number, information entropy and system dynamics integrated model was established to conduct relationship structure simulation and evolution trend evaluation of the WSEE system. The application results in Hefei city, Anhui Province, China, demonstrated that: (1) the variation of overall equilibrium conditions of WSEE system in Hefei city, 2020–2029 was higher compared to that of 2010–2019, though the increasing rate of ordered degree and connection number entropy (ODCNE) became slower after 2019; and (2) the annual ODCNE value from 2020 to 2029 of WSEE system under dry year scenarios increased about 0.0812, which indicated that the construction of Yangtze-Huaihe Diversion (YHD) project could play significant positive role in mitigating the equilibrium situation of WSEE system in Hefei city in the future. On the whole, this study is capable of providing the guidance basis for constructing a theoretical framework of structure simulation and equilibrium evaluation analysis of WSEE complex system.

## 1. Introduction

As an irreplaceable resource for the survival of organisms and humans, water is one of the most fundamental natural resources and strategic economic resources to ensure the high-quality and coordination development of water resources, society, economy and ecological environment (WSEE) complex system [1,2]. Coincident with the rapid development of contemporary society, water resources carrying pressure for increasing population and water demand is continuously aggravated, and this has caused severe water security issues, including water resource shortage, water ecology degradation, water environment pollution and water disaster frequent-occurrence in China [3,4]. It has been estimated that nearly two-thirds of cities in China are confronting different scales of water shortages and even more than 11,000 water-quality-related emergencies have occurred over the last two decades [5,6]. Therefore, given the actual features of large land area, spatio-temporal disequilibrium distribution of water resources and prominent contradiction between water resources supply and demand sides, the Chinese Government released a new strategy framework characterized by water-saving priority, spatial equilibrium, systematic management and implement from two sides of market and government in the field of water resources management in 2014 [7]. This has become a fundamental guideline to coordinate the contradiction between water resources utilization, social economy development and ecological environmental protection.

Equilibrium, originally a core concept from the science of economics, is generally applied to reflect a stable structure achieved through game, and none of the game partners are willing to alter its strategies alone [8]. Spatial equilibrium is widely defined as a regional productivity distribution pattern that satisfies the requirements of sustainable development and coordinates with resources and environmental distribution features [9]. Subsequently, D. H. Bian et al. proposed that water resources spatial equilibrium was primarily focusing on the coordination development of water resources and social economy through water resources regulation and allocation pattern [9,10]. In addition, X. Y. Xu et al. indicated that water resources spatial equilibrium is the extension of water resources carrying capacity, and can be interpreted as the spatial distribution of loadable area interacted between water resources, social economy and ecological environment system [11]. Therefore, it can be indicated that previous research primarily concentrated on the concept exploration and equilibrium evaluation of the WSEE system, but neglected to reveal the mutual feedback relationship and structure features between different elements of the WSEE system. These are all fundamental tasks to simulate the future evolution trend of the WSEE system [1,12], and also the motivation of this study.

To date, many researchers have conducted extensive explorations in terms of comprehensive evaluation and structural simulation of the WSEE system by adopting a variety of approaches, such as synergetics method [13], Gini coefficient and set pair analysis coupling method [14,15], ordered degree [16], system dynamics (SD) [4,17] and an agent-based model (ABM) [18,19]. In particular, H. Gan et al. proposed multi-dimensional normalized evaluation and critical regulation model of a water resources system based on the characteristic exploration of water resources, economy, society, ecology and environment subsystems [13]. Y. F. Yang et al. and C. Dai et al. established connection number, Gini coefficient and variable fuzzy set integrated model to conduct water resources spatial equilibrium evaluation [14,15]. In addition, Y. L. Zhou et al. constructed an adaptive optimal allocation model of a water resources system comprised by multiple modules of multi-agent simulation, multi-objective optimization and system comprehensive evaluation [18]. G. Burger et al. constructed a ABM-SD coupling model to simulate the relationship between water consumption behaviors and land using decisions [20]. On the whole, the abovementioned findings are all beneficial attempts to reveal the relationship structure and evolution mechanism of the WSEE system. Actually, the evolution of the WSEE system is a complicated process accompanied by too many uncertainties; traditional evaluation approaches are unable to describe their influences on the overall evaluation result [6,21]. In addition, the structure simulation analysis of the WSEE system through the SD method from the perspectives of water resources supply and demand balance, as well as the pressure-state-response (PSR) relationship of water resources spatial equilibrium system, is still not reported yet.

In this study, firstly, we primarily applied the connection number and ordered degree methods to describe the membership characteristics between different equilibrium indicators and its belonging evaluation grades. Secondly, the system dynamics (SD) approach was utilized to reveal the feedback relationship features between different indicators and subsystems. Finally, a quantitative equilibrium evaluation and structure simulation model of the WSEE system was proposed, and a combination of ordered degree and information entropy methods was also the primary novelty of this study, which was expected to incorporate the overall influence of uncertainties from the perspectives of identity, difference and opposition. Furthermore, the reliability of the proposed model was verified through its application in equilibrium evaluation during historical years and future evolution simulation of the WSEE system by means of the construction of different scenarios considering the influences of water resources availability, economy development, water saving strategies and water diversion project, etc. Overall, the structure of the manuscript is organized as follows: the background of the study is briefly introduced in Section 1. Section 2 introduces the case study area, the acquisition of related data and research framework. Section 3 establishes the structure simulation and equilibrium evaluation model of the WSEE system coupling system dynamics, connection number and ordered degree methods. Section 4 discusses the historical variation characteristics, future evolution trend under different development scenarios of WSEE system in Hefei city China, and policy implications and limitations of this study, and Section 5 presents the main conclusions.

## 2. Study Area and Research Framework

### 2.1. Study Area and Data Sources

Hefei city, located between 30°57′–32°32′ N and E 116°41′–117°58′, is the capital of Anhui province in the Yangtze River Delta and hinterland of east China (as indicated in Figure 1). Hefei city is also located in the north subtropical and humid climate zone, which has jurisdiction over four districts, four counties and one city including Yaohai, Luyang, Baohe, Feixi, Chaohu city etc., and covers a total area of 11,445 km^2^. The economy of Hefei city has developed rapidly during the recent ten years, and the GDP of Hefei city has reached about 140 billion$ in 2020, which is approximately 2.7 times compared to 2011. In addition, 82 rivers with catchment area larger than 50 km^2^, belonging to two main river basins separately, flows across Hefei city, i.e., about 78% of the city area (8824 km^2^) belongs to Yangtze River basin and 22% of the city area (2606 km^2^) belongs to Huaihe River basin. Moreover, the primary rivers and lakes of Hefei city includes Chaohu Lake, Wabu Lake, Nanfei River etc., and Chaohu Lake, with a total watershed area of 13,486 km^2^, is one of the top five biggest freshwater lakes in China. Moreover, the average annual water resources in Hefei city is 4.78 billion m^3^ from 2010 to 2020, which is higher than the average of 2.04 billion m^3^ from 2000 to 2009 because of the optimal utilization of water conservancy projects. However, the total water resources utilization of Hefei city in 2020 has reached to 3.09 billion m^3^, and the distribution of surface water resources in Hefei city presents remarkable decreasing trend from south to north. It has been investigated that the total water resources utilization of cities between Yangtze and Huaihe rivers presented obvious increasing trend since 2000, and the annual increasing rate has approached to about 2.92%, which is even higher in domestic water and industrial sectors with approximately 4.8%.

Therefore, to effectively mitigate water resources supply pressure for the cities of Huaihe river basin, the Yangtze-Huaihe Diversion (YHD) project was constructed by Chinese government. As shown in Figure 1, the YHD project can be divided into three parts from south to north, including Yangtze-Chaohu Diversion project, Yangtze-Huaihe Connection project and Yangtze Water Diversion to North project, firstly, Zongyang hub and Fenghuangjin hub projects on the left side of main stream of Yangtze river are utilized as the water intake junction, and then water flows into Chaohu lake along Caizi lake and Xizhaohe river route, and then flows into the main stream of Huaihe river through Wabu lake ultimately. According to the future water diversion planning schemes of YHD project, it is estimated that the added water supply amount through YHD project to Hefei City will reach to 0.031, 0.069 and 0.183 billion km^3^ under three water resources availability scenarios of wet year, normal year and dry year by 2030.

In brief, the distribution of water resources, population density and social economy development among production, domestic and ecological sectors exist evident disequilibrium characteristics in water-receiving area of YHD project, which makes it urgent to conduct equilibrium evaluation and structure simulation of the WSEE system. Therefore, we collected the historical statistic data of Hefei city and other regions of water-receiving area of YHD project from Water Resources Bulletin of Anhui Province, Statistical Yearbook of Anhui Province, EPS statistic data platform (https://www.epsnet.com.cn/index.html#/Index accessed on 20 May 2022) and China’s economic and social big data research platform (https://data.cnki.net/ accessed on 20 May 2022), and the related research finding is excepted to be beneficial to establish regional water resources development and utilization schemes and strategies.

### 2.2. Research Framework

Overall, the structure for the contents of this manuscript can be built as follows: (1) the primary task of Section 1, methodologies, is to establish the SD method based model for relationship structure simulation and Ordered Degree and Connection Number Entropy (ODCNE) coupling model for comprehensive assessment analysis of the WSEE system. The essence for the establishment of the structure simulation model is describing and revealing the water supply-demand balance and PSR relationships between water resources supply and utilization, social economy and ecological environment subsystems through the SD approach. The key issues for the development of a comprehensive evaluation model includes the calculation of ODCNE value of single indicators and the derivation of overall equilibrium contribution value of evaluation samples through weight series of indicators; (2) in terms of Section 2, the reliability of the proposed approach was further verified through its application in terms of simulation error analysis, comprehensive evaluation analysis of historical samples, and variation trend prediction analysis of the WSEE system under different scenarios in the future. In addition, the adaptive regulation policy and suggestions, limitations and future work were also discussed. The explicit relationship and details of the abovementioned modules was illustrated in Figure 2.

## 3. Methodologies

### 3.1. Structure Simulation of WSEE System by SD Model

System Dynamics (SD) method, proposed by Prof. J. W. Forrester from MIT in 1956, is a well-established system simulation approach for understanding, visualizing and analyzing complex dynamic feedback and relationship structure which exhibits nonlinear, multi-feedback and time-varying properties among different elements and subsystems [6,22]. Compared with other commonly applied system simulation methods, SD model is capable of addressing systematic problems through microscale simulation of internal dynamic relationship and feedback mechanism as well as macroscale regulation of external function structure [6,18]. Briefly, the basic building elements of SD method are state variable (S), rate variable (R), auxiliary variable (A) and information arrow, as shown in Figure 3, in which, state variables (S) represent the cumulative effect of the system information over time, rate variables (R) reflect the speed of system information cumulative effect and the changes of state variables over time, and auxiliary variables (A) are intermediate variables which run through the system changing and decision-making process, and indicate the existence of a certain relationship between two variables linked by an information arrow [23].

In this study, we applied the SD model to simulate the complex feedback relationship and structure of the WSEE system involving water resources, population, society, economy and environment factors, such as the supply-demand balance relationship between water resources supply and utilization sectors as well as PSR relationship between water resources, social economy and ecological environment subsystems, so as to reveal the external function and future variation trend of the WSEE system. In addition, the SD model was developed through multiple detailed parameter and numerical equation estimations, which was aiming to reduce uncertainties and improve simulation accuracy of the model. Moreover, the WSEE system is typically influenced by many variables defined by connections, feedback and restrictions among them, therefore, based on the water resources supply-demand balance and fundamental carrying characteristics within the WSEE system described above as well as the actual society development situation, the WSEE system was divided into four subsystems in this study, including social economy subsystem, water resources supply subsystem, water resources utilization subsystem and ecological environment subsystem. The Vensim PLE software tool was applied to exhibit the main feedback and connections of different variables and subsystems, as indicated in Figure 4, and all the detailed definitions and equations of different variables used in this study can be referred in Appendix A.

#### 3.1.1. Social Economy Subsystem

The social economy subsystem is a significant component of the WSEE system and has substantial impacts on the variation of other subsystems [6]. As indicated in Figure 4a, the performance of social economy subsystem itself and its influence on the overall variation of the WSEE system were primarily explored in terms of population increase and economy output [6,23,24]. The total population is mainly determined by natural growing rates of urban and rural population, and some auxiliary variables were introduced to study the influence of population increase on the overall development of the WSEE system [6,25]. In terms of economy output, the GDP constitutes the core indicator of economy accounting and represents the variation of economy output [6,26]. Thus, regional GDP is the most significant variable of social economy subsystem, which was remarkable influenced by multiple auxiliary variables of natural growing rate of urban population, ten thousand Yuan industrial added value and water resources supply price. On the whole, the primary controlling indicates dominating the variation of social economy subsystem includes urbanization rate (UR), GDP per capital (GDPC) and population density (PD), which were also applied as evaluation indicators of the WSEE system in Section 3.2 and have been colored in red in Figure 4a.

#### 3.1.2. Water Resources Supply Subsystem

Water resources supply subsystem is the foundation for the sustainable development of the WSEE system. It provides essential water resources required by population to meet basic domestic water requirements, industrial sectors to satisfy water utilization requirements of production output, agricultural irrigation sectors to ensure normal growth of crops, and also ecological sectors to sustain the healthy development of water environment system [6,18]. Reversely, the above water resources utilization subsystems (i.e., domestic, industrial, agricultural and environmental sectors) also have negative impacts on the water resources supply subsystem; excessive increases of water resources demand will undoubtedly aggravate water resources supply burden, and even result in an imbalanced relationship between water resources supply and utilization subsystems [27]. As indicated in Figure 4b, the internal structure of water resources supply subsystem comprised three parts, including surface water resources (SWRA), groundwater resources (GWRA) and water diversion amount through YHD project (WDAH). In addition, water resources supply subsystem also has feedback relationship with other elements of the WSEE system, such as water resources utilization, population, land area, etc. The dominating indicators of water resources supply subsystem includes water resources availability per capita (WRAC), modulus of water resources production (MWAP) and rate of water resources development and utilization (RWDU), which have been colored in red in Figure 4b and also selected as evaluation indices of the WSEE system in Section 3.2.

#### 3.1.3. Water Resources Utilization Subsystem

Water resources utilization subsystem forms the other core component of the entire WSEE system, which includes five parts in this study, i.e., domestic water consumption (DWC), industrial water consumption (IWC), agricultural irrigation water consumption (AIWC), tertiary industrial water consumption (WCTI) and ecological water consumption (EWC), as indicated in Figure 4c. Specifically, DWC reflects the quantity of water resources utilized for population’s daily life, which was determined through daily domestic water consumption per capital (DDWCC) and total population for both urban and rural regions. IWC refers to the portion of water resources consumption for the purpose of social production output, which was mainly determined by total industrial output and corresponding water consumption efficiency. AIWC primarily refers to the water consumed to meet the agricultural irrigation requirements of crops, which accounts for the largest portion of water consumption and dominantly determined through quota of agricultural irrigation (QAI) and total irrigation area (IA) in this study. EWC is the quantity of the water resources consumed by green areas, environmental sanitation and surface evaporation, which was also simply calculated through the quota of urban green land area irrigation (QUGLA) and total urban green land area (UGLA) in this study. Likewise, several important auxiliary variables, such as DDWCC, water consumption per capital (WCPC), QAI and water consumption of ten thousand Yuan GDP (WCTYG), have also been selected as comprehensive evaluation indices of the WSEE system in Section 3.2, which were colored in red in Figure 4c.

#### 3.1.4. Ecological Environment Subsystem

Generally, the quality of the ecological environment will reflect the sustainability of urban development, and is closely related to the residential environment, industrial development, as well as human health [28]. As indicated in Figure 4d, the total ecological water consumption will grow with the variation of urbanization rate (UR), the increasing rate of urban green land area (IRGLA), vegetation coverage rate (VCR) and rate of urban sewage treatment (RUST) [6,29], and to a certain extent, the RUST index can be improved by increasing the investment of sewage management projects. Similarly, several controlling variables of ecological environment subsystem, such as VCR, RUST and the rate of ecological water consumption (REWC), were also identified as the evaluation indicators to represent the overall variation of the WSEE system in Section 3.2, which were colored in red in Figure 4d.

### 3.2. Equilibrium Evaluation of WSEE System by ODCNE Model

#### 3.2.1. Evaluation Index System of WSEE System

The development of an evaluation index system is fundamental for exploring the relationship structure characteristics of the WSEE system. Firstly, from the perspective of system structure characteristics of “pressure-state-response” and combining the principles of evaluation indicator selection including scientificity, completeness, comparability as well as operability: three indicators (S1–S3) were selected to reveal the fundamental carrying function of the water resources system to the development of society and economy; four indicators (S4–S7) of the society system; three indicators (S8–S10) of the economy system were chosen to represent the pressure effect to water resources system; and three indicators (S11–S13) were applied to describe the sustainable situation for the development of the ecological environment system. Secondly, according to the feedback relationship between pressure indicators (negative) and carrying indicator (positive) of the WSEE system, the variation of different indicators was divided into three levels including Grade I (loadable), Grade II (critical) and Grade III (overloaded), and the indicator threshold value corresponding to different variation grade were also obtained through expert experience and social investigation results. Finally, the comprehensive evaluation index system of the WSEE system comprising four subsystems of water resources, society, economy and ecological environment, 13 equilibrium indicators and three equilibrium grades, was established in this manuscript [30,31], which was shown in Table 1. In addition, the weight vector corresponding to different indicators was also determined through the Accelerating Genetic Algorithm-Fuzzy Analytical Hierarchy Process (AGA-FAHP) method, and the detailed calculation procedures of AGA-FAHP can be referred from reference [32].

#### 3.2.2. Connection Number of Set Pair Analysis Method

Set Pair Analysis (SPA), proposed by Chinese scholar K. Q. Zhao in 1989, was a quantitative analysis method to describe the transforming characteristics from uncertainty to certainties of complex system [33,34]. Set pair is defined as two set with certain connection attributes, then any two elements within the system can be regarded as set pair from the perspective of systematic science [15,33]. The essence of SPA is regarding the two sets related to the research objectives as a complex system filling with uncertainty and certainty features, and then discussing the connection attributes between different samples, indicators and criteria system from the perspectives of identity, difference and opposition, which is excepted to reveal the complex fuzzy relationships within the complicated system more elaborately and rationally in comparative with tradition systematic analysis approaches [15,33,34]. The quantification of fuzzy connection relationship of complex set pair system through SPA is frequently accomplished by the determination of connection number, as follows:
(1)u=a+b⋅I+c⋅J
where *u* is connection number of set pair system, *a*, *b* and *c* are the identical degree, difference degree and opposition degree of the set pair system, respectively; *I* is difference degree coefficient, varying within [−1, 1], and *J* is opposition degree coefficient, equaling to −1 constantly. Essentially, connection number is applied to describe the combination situation of three driving elements including identity, difference and opposition within set pair system in a certain period, and it is exactly the feedback relationship between these three elements that promotes the continuous variation of the set pair system. In addition, taking the connection number concept as the core, the derived concept of connection entropy, subtraction set pair potential and connection cloud model etc., can be also utilized to reveal the transformation features of identity, difference and opposition features [35], which have been widely applied in the fields of groundwater dynamic prediction [36], water resources carrying capacity assessment and trend analysis [15,37], and water environment quality evaluation [30], etc.

#### 3.2.3. Information Entropy and Ordered Degree Approaches

The WSEE system is a typical uncertain system which exists continuously in information exchange with the external environment. Entropy is essentially an effective tool to quantitatively reveal the uncertainties throughout the evolution of the WSEE system [38,39]. Subsequently, entropy was extended as a quantitative indicator to measure the disorder conditions of the system, and the higher the entropy value, the more ordering of the system structure. Essentially, every natural system can be regarded as a dissipative structure system, and also characterized with three features [40,41]: (1) the distribution of energy and resources within the system is disequilibrium; (2) the system is open and can exchange information with external environment; and (3) the nonlinear relationship does exist between system elements [38,42]. Reversely, if the variation of system structure is satisfied with the above three requirements, the system ordered degree will increase gradually through interaction and feedback regulation within the system elements to formulate new dynamically ordered structure [42]. Generally, the WSEE system is a typical dissipative structure system with disequilibrium, open and self-organizing characteristics, and is also constrained by water resources availability, ecological environment and social economy development [16,40,41,42]. Therefore, the information entropy and three-element connection number methods were utilized to describe the varying features of different equilibrium indicators of the WSEE system, and the Ordered Degree and Connection Number Entropy (ODCNE) was proposed to describe the overall equilibrium situation of the WSEE system in this manuscript, the more higher the ODCNE value. The higher the equilibrium degree between different indicators, the more stable the equilibrium structure of the WSEE system would be.

#### 3.2.4. Ordered Degree and Connection Number Entropy (ODCNE) Coupling Model

In this section, based on the summarizing of previous research results related to the connotation characteristic and relationship structure of the WSEE system, the comprehensive equilibrium effect of the WSEE system was quantitatively described by means of the feedback relationship between the fundamental carrying function of water resources system and the pressure demand of society, economy and environment system. Additionally, the ordering characteristic value of different indicators and subsystems was applied to represent the overall equilibrium situation of the WSEE system. Specifically, the three-element connection number method was applied to describe the single-indicator connection attribute between different equilibrium index and its corresponding varying grade, and the ODCNE was utilized as the ultimate measuring index to represent the equilibrium degree of different evaluation indicators of the WSEE system; in other words, the ODCNE could be obtained through four steps, as follows.

***Step*** 1: the determination of three-element connection number of single indicators of the WSEE system. Based on the establishment of evaluation indicator and grade division standard of the WSEE system, if regarding the indicator value *x_ijk_* of *i*th sample, *j*th subsystem and *k*th indicator and varying grade threshold value *s_gj_* of *g*th level as a set pair system, then the corresponding single-indicator and three-element connection number *v_ijkg_,* varying within [−1, 1], could be determined through SPA method [15,30,35], as follows:
(2)$$v_{ijk1}=\left\{\begin{array}{c} 1,\;\mathrm{positive}\;\mathrm{indicator}\;x_{ijk}\leq s_{1k}\;\mathrm{or}\;\mathrm{negative}\;\mathrm{indicator}\;x_{ijk}\geq s_{1k}\\ 1-2(x_{ijk}-s_{1k})/(s_{2k}-s_{1k}),\;\mathrm{positive}\;\mathrm{indicator}\;s_{1k}<x_{ijk}\leq s_{2k}\;\mathrm{or}\;\mathrm{negative}\;\mathrm{indicator}\;s_{1k}>x_{ijk}\geq s_{2k}\\ 1,\;\mathrm{positive}\;\mathrm{indicator}\;x_{ijk}>s_{2k}\;\mathrm{or}\;\mathrm{negative}\;\mathrm{indicator}\;x_{ijk}<s_{2k} \end{array}\right.$$
(3)$$v_{ijk2}=\left\{\begin{array}{c} 1-2(s_{1k}-x_{ijk})/(s_{1k}-s_{0k}),\;\mathrm{positive}\;\mathrm{indicator}x_{ijk}\leq s_{1k}\;\mathrm{or}\;\mathrm{negative}\;\mathrm{indicator}\;x_{ijk}\geq s_{1k}\\ 1,\;\mathrm{positive}\;\mathrm{indicator}s_{1j}<x_{ijk}\leq s_{2j}\;\mathrm{or}\;\mathrm{negative}\;\mathrm{indicator}\;s_{1k}>x_{ijk}\geq s_{2k}\\ 1-2(x_{ijk}-s_{2k})/(s_{3k}-s_{2k}),\;\mathrm{positive}\;\mathrm{indicator}s_{2k}<x_{ijk}\leq s_{3k}\;\mathrm{or}\;\mathrm{negative}\;\mathrm{indicator}\;s_{2k}>x_{ijk}\geq s_{3k}\\ -1,\;\mathrm{positive}\;\mathrm{indicator}x_{ijk}>s_{3k}\;\mathrm{or}\;\mathrm{negative}\;\mathrm{indicator}\;x_{ijk}<s_{3k} \end{array}\right.$$
(4)$$v_{ijk3}=\left\{\begin{array}{c} -1,\;\mathrm{positive}\;\mathrm{indicator}\;x_{ijk}\leq s_{1k}\;\mathrm{or}\;\mathrm{negative}\;\mathrm{indicator}\;x_{ijk}\geq s_{1k}\\ 1-2(s_{2k}-x_{ijk})/(s_{2k}-s_{1k}),\;\mathrm{positive}\;\mathrm{indicator}\;s_{1k}<x_{ijk}\leq s_{2k}\;\mathrm{or}\;\mathrm{negative}\;\mathrm{indicator}\;s_{1k}>x_{ijk}\geq s_{2k}\\ 1,\;\mathrm{positive}\;\mathrm{indicator}\;s_{2k}<x_{ijk}\leq s_{3k}\;\mathrm{or}\;\mathrm{negative}\;\mathrm{indicator}\;s_{2k}>x_{ijk}\geq s_{3k} \end{array}\right.$$
where *s*_0*j*_ and *s*_3*j*_ are the left and right varying threshold values of *j*th indicator, respectively, *s*_1*j*_ is the critical threshold value of grade I and II of *j*th indicator, and *s*_2*j*_ is the critical threshold value of grade II and III of *j*th indicator. Meanwhile, the higher of the positive indicator value, the higher of its corresponding equilibrium grade, which is exactly opposite for negative indicator.

***Step*** 2: the determination of the component of ODCNE *u_ijk_* corresponding to *i*th sample, *j*th subsystem and *k*th indicator. The relative membership degree *v*_ijkg_* of indicator value *x_ijk_* belonging to fuzzy concept of *g*th grade could be calculated [30], as follows:
(5)$${v^{*}}_{ijkg}=0.5+0.5\cdot v_{ijkg}\;(i=1\sim M;\;j=1\sim N;\;k=1\sim K;\;g=1\sim G)$$

Then, the normalized connection number component *v_ijkg_* of indicator value *x_ijk_* belonging to fuzzy concept of *g*th equilibrium grade, represented from the perspective of identity, difference and opposition, was determined through normalization processing of relative membership degree *v*_ijkg_* obtained by Equation (6) [15,30,35], as follows:
(6)$$v_{ijkg}=\frac{{v^{*}}_{ijkg}}{\sum_{g=1}^{G}{v^{*}}_{ijkg}}$$

The single-indicator and three-element connection number *u*_ijk_* was determined through the integration of three component values *v_ijk1_*, *v_ijk2_*, *v_ijk3_* [30], as follows:
(7)$${u^{*}}_{ijk}=v_{ijk1}+v_{ijk2}\cdot I+v_{ijk3}\cdot J$$
where the difference degree coefficient I was determined basing on proportion division of identity, difference and opposition, i.e., $I=(v_{ijk1}-v_{ijk3})+\frac{v_{ijk1}-v_{ijk3}}{v_{ijk2}}$, and opposition degree coefficient J was constant, equaling to −1 [15,37]. Thus, the overall connection number *u*_ijk_* was converted to the relative membership degree *u_ijk_*, which represented the coordinating variation situation of *i*th sample, *j*th subsystem and *k*th indicator, as follows:
(8)$$u_{ijk}=0.5+0.5\cdot {u^{*}}_{ijk}\;(i=1\sim M;\;j=1\sim N;\;k=1\sim K)$$

***Step*** 3: the determination of ordered degree *y_ij_* indicating the contribution of variation of *j*th subsystem to the overall equilibrium evolution of *i*th sample. The contribution of subsystems of water resources, society, economy and ecological environment to the overall equilibrium evolution situation of *i*th evaluation sample was quantified through the integrating calculation of ordered degree *y_ij_* by weight vector [15], as follows:
(9)$$y_{ij}=\sum_{k=1}^{K}\omega_{k}\cdot u_{ijk}$$
where *y_ij_* was ordered degree corresponding to *i*th sample and *j*th subsystem, *w_k_* was the weight value of *k*th indicator.

***Step*** 4: the determination of ODCNE value *S_i_* representing the overall equilibrium variation of *i*th sample. In combination with the information entropy concept, the ODCNE value *S_i_* of *i*th sample was utilized to quantitatively represent the overall variation trend as well as its belonging equilibrium grade in this study [15], as follows:
(10)$$S_{i}=-\frac{1}{\ln N}\sum_{j=1}^{N}y_{ij}\cdot \ln y_{ij}$$

## 4. Results and Discussion

### 4.1. Error Analysis of Structure Simulation of WSEE System through SD Model

A simulation model is a system that abstracts the real world into a relationship structure, and whether it is able to accurately represent the practical system depends on the degrees of simulation error [43]. Therefore, in this section, we firstly utilized the historical data of different equilibrium evaluation indicators of Hefei city, 2010–2019, to verify the validity of the SD model for structure simulation of the WSEE system. In detail, the historical data of different indicators from 2010 to 2019 were fed into the model, and then the simulation results were derived through parameters estimation and also compared with the observed values. Due to the large amounts of calculation in the model, only four representative equilibrium indicators corresponding to four different equilibrium subsystems were selected to testify the simulation accuracy, i.e., the rate of water resources development and utilization (RWDU), population density (PD), the water consumption of ten thousand Yuan GDP (WCTYG) and rate of ecological water consumption (REWC), and the comparative results of the four typical indicators were shown in Figure 5. Meanwhile, we also applied the Accumulated Relative Error Value (AREV) and Correlation Coefficient Value (CCV) of the entire 13 equilibrium indicators (as shown in Table 1) to further verify the effectiveness of the SD model for structure simulation of WSEE system, which was illustrated in Figure 6.

It can be revealed from Figure 5 and Figure 6 that: (1) the simulation result of the four representative indicators of RWDU, PD, WCTYG and REWC all matched well with the actual values with the CCV exceeding 0.81, and the average of CCV for entire 13 indicators was 0.78, which falls within the acceptable range of correlation verification; (2) the top four indicators with the highest AREV are RWDU, DDWCC, WCPC and WCTYG with the absolute value of AREV all exceeding 0.85, and the average of AREV for entire 13 indicators of the WSEE system was −0.0218; and (3) the comparative results of CCV and AERV all revealed that the structure simulation model based on SD method effectively reflected the reality and could provide a good foundation for the evolution trend and development scenarios analysis of the WSEE system.

### 4.2. Equilibrium Evaluation Analysis of WSEE System

Besides the relationship structure simulation of the WSEE system through the SD method, another objective of this study is to comprehensively evaluate the annual equilibrium state of the WSEE system during historical years and also explore its varying trend in future. Therefore, the ODCNE data series representing the overall equilibrium state of the WSEE system from 2010 to 2019 in Hefei city was obtained through the proposed connection number and ordered degree entropy coupling model. Furthermore, we predicted the values of different equilibrium indicators of the WSEE system from 2020 to 2029 in Hefei city through the proposed structure simulation model utilizing the same estimated values of auxiliary variables and rate parameters of SD method as the years from 2010 to 2019, and then the variation of ODCNE value from 2020 to 2029 can be also determined through the proposed equilibrium evaluation model. Ultimately, the overall variation of ODCNE from 2010 to 2029 in Hefei city was shown in Figure 7.

It can be concluded from Figure 7 that: (1) the variation of ODCNE of the WSEE system presented remarkable increasing trend from 2010 to 2029, indicating that the overall equilibrium situation between water resources, society, economy and ecological environment subsystems will be improved significantly after 2010; (2) comparatively, the average of ODCNE increased from 0.8784 during 2010 to 2019 to 1.0118 during 2020 to 2029, which revealed that the overall equilibrium state of the WSEE system from 2020 to 2029 in Hefei city was higher than before, while the increasing rate of ODCNE became slower after 2019; (3) comparing with the years from 2020 to 2029, the fluctuation of ODCNE was evident during the historical years from 2010 to 2019, and the highest and lowest values of ODCNE occurred in 2014 and 2011, respectively, which was basically consistent with the variation trend of indicators including water resources availability per capita (WRAC), water consumption per capital (WCPC) and modulus of water resources production (MWRP); and (4) total water resources amount (TWRA) and water resources consumption (WRC) are the two primary indicators which have dominate impacts on the variation of ODCNE of WSEE system, the values of TWRA and WRC of Hefei City in 2011 were 2.964 and 3.228 billion m^3^ separately, which was almost the peak among the historical years and also consistent with the variation of ODCNE, and this was exactly the opposite with the variation in 2014. Accordingly, the variation trend of WRAC, WCPC and MWRP were basically the same as that of TWRA and WRC during historical years. Therefore, it is quite necessary to increase water resources availability through effective strategies, such as water diversion project, reservoir regulation, water-saving scheme implement etc., to further improve overall equilibrium situation of the WSEE system in Hefei city especially during dry years in future.

### 4.3. Equilibrium Evaluation under Different Future Scenarios of WSEE System

#### 4.3.1. Scenario Schemes Management

As mentioned above, the ultimate objective of this manuscript is to explore the variation of equilibrium situation of the WSEE system in Hefei city under different future development scenarios. For this purpose, we chose different scales of water resources availability, water-saving scheme implement and economy development to represent different scenarios of water resources supply, water resources utilization and social economy development subsystems in the future. Specifically, annual precipitation, which was classified into three levels including wet year, normal year and dry year, was regarded as the primary scenario parameter of water resources supply subsystem. The threshold values of annual precipitation corresponding to different levels was determined through precipitation frequency analysis during historical years in Hefei city. In addition, the degrees of water-saving scheme implement and economy development were classified into two levels (high and low), and the related values and calculation equation of different indicators under different levels were obtained according to parameter experiential estimation (QAI, WCTIAV), regression analysis (DDWCC, WCTI, TIAV and GDP) and government planning report 2020 (UPGR, industrial water utilization price, https://www.hefei.gov.cn/public/1741/106913659.html accessed on 20 May 2022). The detailed values and calculation equations of different indicators were shown in Table 2.

Subsequently, basing on the estimated values and calculation equations of different indicators under different development scenarios listed in Table 2, 12 simulation schemes of the WSEE system in Hefei city considering different development scenarios of water resources supply, utilization and social economy development subsystems were provided through indicators combination, as illustrated in Table 3.

#### 4.3.2. Equilibrium Evaluation Analysis under Different Simulation Scenarios

Based on the establishment of simulation scheme set considering different development scenarios, the variation of ODCNE representing the overall equilibrium situation of the WSEE system from 2020 to 2029 in Hefei city can be obtained through the proposed equilibrium evaluation model, as indicated in Figure 8.

In addition, the average of ODCNE of different schemes under different development scenarios in the future was indicated in Table 4.

It can be concluded from Figure 8 and Table 4 that:
(1)The simulation scheme of the WSEE system, 2020–2029, was obtained utilizing the same estimated values of auxiliary variables of SD model as that of the years from 2010 to 2019 (as mentioned in Section 4.2). If taking the simulation scheme of the WSEE system as the basic scheme, then the average of ODCNE of schemes 1#, 2#, 3# and 4# under dry year scenario were 0.8674, 0.9736, 0.8754 and 1.0278, respectively, which were almost all lower than the average of basic scheme (1.0113), and this was exactly opposite with the schemes 5#, 6#, 7# and 8# under wet year scenario. In terms of normal year scenario, the average of ODCNE of schemes 9# and 11# under low water saving scenario were 0.9456 and 0.9493 separately, which were all lower than the average of basic scheme.(2)As compared to the influences of water resources availability factor, the average of ODCNE for the schemes with high economy development and water saving measures under same water resources supply scenario are higher than that of the schemes with low economy development and water saving measures. It was estimated that the variation scale of ODCNE caused by economy development and water resources supply factors under high water saving scenarios was approximately 0.0589, while accordingly, the variation scale of ODCNE under low water saving scenarios was negligible.(3)It was evident that the impacts of water resources availability on the variation of overall equilibrium state of the WSEE system is the most serious, and followed by water saving factor of water resources utilization scenario, and the influences of society development factor under low water resources utilization scenarios can be even negligible. Therefore, it could be essential to improve the water resources supply situation through water resources diversion project and reservoir regulation, etc., so as to enhance the overall equilibrium state of the WSEE system, especially during dry years.

Overall, if comprehensively considering the influences of economy development and water saving implement, the average of ODCNE under a dry year scenario in the future was 0.9361, which was all lower compared to both wet and normal year scenario. This exactly reflects the necessities for the construction of YDH Project to regulate the equilibrium situation of the WSEE system especially during future dry years, and thus it is also crucial to quantitatively discuss the influences of YDH Project to the entire WSEE system combining the future economy development and water saving implement factors.

#### 4.3.3. Impacts Analysis of YHD Project under Dry Year Scenarios

In this section, we further discussed the positive influence of the YHD project to the overall equilibrium variation of the WSEE system. It can be estimated that the amount of water resources diversion through the YHD project to Hefei city in 2030 under dry year scenarios is approximately 0.5889 billion m^3^ according to the planning report of YHD project, 2020–2030, and then the total water resources availability in Hefei city, 2020–2030, under dry year scenarios could be obtained as well through the SD based simulation model of water resources supply subsystem (indicated in Figure 4b). Therefore, the variation of ODCNE of different schemes under dry year scenarios, i.e., considering the influences of the YHD project, could be eventually obtained, as shown in Figure 9.

It can be revealed from Figure 9 that: (1) the equilibrium condition of the WSEE system in Hefei city under dry year scenarios was remarkably improved because of the beneficial influence of YHD project implement. In detail, the average of ODCNE of schemes 1–4# under future dry year scenarios was 0.8637, 0.9602, 0.8702 and 1.0088, respectively, which were almost all lower than the simulation scheme (as shown by red solid line). After the construction of the YDH project, the corresponding average value of ODCNE of schemes 1–4# increased to 1.0088 to 0.9425, 1.0361, 0.9439 and 1.0763, respectively, which was almost approaching to the equilibrium state under future normal year scenario; (2) on the whole, the annual ODCNE of the WSEE system from 2020 to 2029 under future dry year scenario increased about 0.0812 as compared to before the construction of the YHD project. In addition, the fluctuation for the variation of ODCNE after the YHD project implementation became lower, which all indicated that the construction of the YHD project could play significant positive role in mitigating the equilibrium state of the WSEE system in Hefei city in the future.

### 4.4. Policy Implications

The ultimate purpose of the structure simulation and equilibrium evaluation analysis of the WESS system in this study is to provide an effective decision-making basis for the establishment of policy strategies and schemes under different future scenarios. Generally, it can be concluded that the impact of water resources availability on the equilibrium situation of the WSEE system is the most significant, followed by water saving policy implementation and social economy development scale. The demand for water resources will increase as the economy and population continue to grow. Under such conditions, more efficiently regulating the mutual feedback relationships between water resources, society, economy and ecological environment systems is tremendously favorable to improve the overall equilibrium situation of the complex WSEE system. Therefore, inspired by the equilibrium evaluation and structure simulation analysis of the WSEE system above, firstly, what we need to do urgently is to increase total water resources availability and enhance its carrying capacity to social economy, population and ecological environment systems through different engineering and non-engineering measures. Secondly, the implementation of water-saving-type society construction and water-saving appliances promotion is another effective countermeasure to enhance water resources utilization efficiency, so as to mitigate the carrying pressure to water resources system. Thirdly, the optimization of industrial layout and water consumption structure are also indispensable. In this case, the increasingly aggravating water resources utilization challenge could be alleviated to a large extent and the water resources development and utilization objective of spatiotemporal equilibrium could be achieved as well.

## 5. Conclusions

In this study, according to the characteristics of the WSEE system in Hefei city, Anhui Province, China, a comprehensive evaluation system incorporating 13 individual indicators was established through the ordered degree and connection number entropy coupling model. Besides, a SD model for simulating the water supply/demand balance and PSR mutual feedback relationships among three equilibrium subsystems was also constructed. After performing a validity test analysis, the application results demonstrated that the proposed approach can adequately capture the essence of the WSEE complicated system in Hefei city. Afterwards, we formulated 12 scenarios incorporating different water resources supply, water resources utilization and social economy development modes in the future to simulate the dynamic variation trend of the WSEE system. After performing calculations, analysis and comparisons of the scenario simulation results, we recognized the primary influencing factors of the WSEE system and provided corresponding regulating suggestions to maintaining the equilibrium development of the WSEE system. In conclusion, the primary findings of this study can be summarized as follows:
(1)The variation of ODCNE of the WSEE system in Hefei city presented a distinct increasing trend through 2010 to 2029, and the highest and lowest values of ODCNE occurred in 2014 and 2011, respectively, during the historical years, which was basically consistent with the variation trend of indicators including WRAC, WCPC and MWRP. In addition, TWRA and WRC are the two primary indicators which have dominate impacts on the variation of ODCNE of the WSEE system, and it is quite necessary to increase the water resources availability through effective strategies to further improve the overall equilibrium situation of the WSEE system during dry years in the future.(2)It was evident that the impacts of water resources availability on the variation of overall equilibrium state of the WSEE system is the most serious, and followed by water-saving factor of water resources utilization scenario. The construction of the YHD project could play significant positive role in mitigating the equilibrium state of the WSEE system in the future, and the equilibrium condition of the WSEE system under dry year scenarios can be tremendously improved to the equilibrium varying state under normal year scenarios. Meanwhile, the integrated simulation and evaluation analysis results of the WSEE system will provide reasonable decision-making basis for the development of annual water resource allocation schemes of the YHD project.

The ultimate motivation of the study is to provide a reliable decision-making basis for the establishment of water resources supply and utilization schemes under different future scenarios. According to the future equilibrium evaluation analysis and influencing priorities of different subsystems and indicators, firstly, what we need to do crucially and effectively to improve the equilibrium situation of the WESE system to optimally regulate water resources supply structure through water diversion project construction, optimal operation of water conservancy projects, etc. In addition, the structure simulation and equilibrium evaluation results of the WSEE system, 2020–2030, can be also beneficial for the establishment of practical future development schemes of social economy, population, water resources allocation, etc. Secondly, the implementation of water-saving-type society construction and water-saving appliances promotion is another effective countermeasure to enhance water resources utilization efficiency so as to mitigate the carrying pressure to water resources system. Thirdly, the optimization of industrial layout and water consumption structure are also indispensable. In this case, the increasingly aggravating water resources utilization challenge could be alleviated to a large extent and the water resources development and utilization objective of spatiotemporal equilibrium could be achieved as well.

All in all, the exploration of this study is crucial preliminary work to realize the sustainable and equilibrium development of the WSEE complex system. However, it is obvious that the SD based model for structure simulation of the WESS system was proposed on the premise of empirical hypothesis of statistical rules of auxiliary variables and correlation equations; how to reveal the internal response mechanism between different variables and relationships is still unsolved. In addition, how to construct reasonable future development scenarios and conduct an effective and quantitative expression of the nonlinear feedback relationships between different equilibrium subsystem and indicators [44] are also limitations and a big challenge, which is also the future research direction of this study.

## Figures and Tables

**Figure 1 entropy-25-00181-f001:**
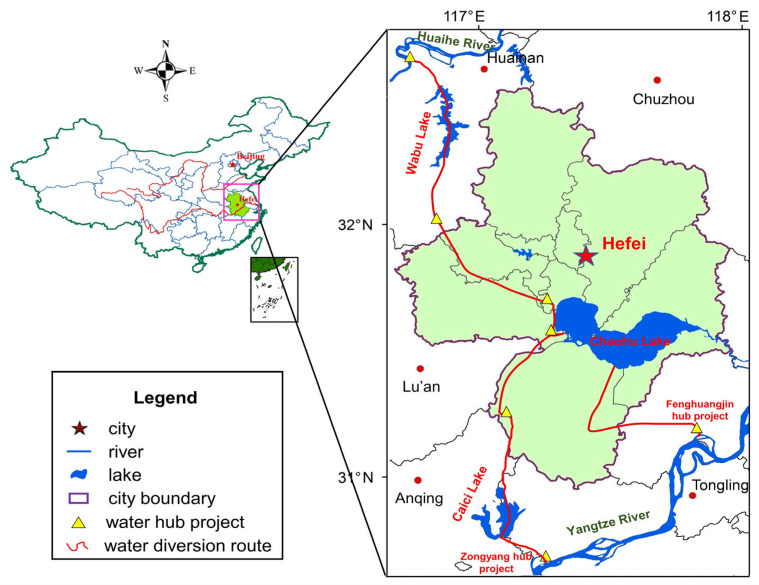
Geographical location and overview of Hefei city in Anhui Province.

**Figure 2 entropy-25-00181-f002:**
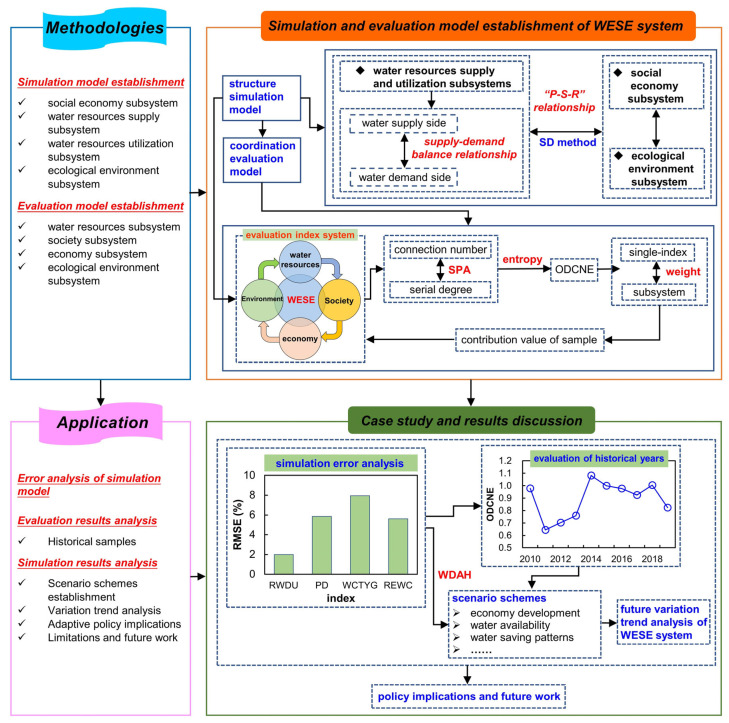
Framework for the simulation and evaluation approach of WSEE system.

**Figure 3 entropy-25-00181-f003:**
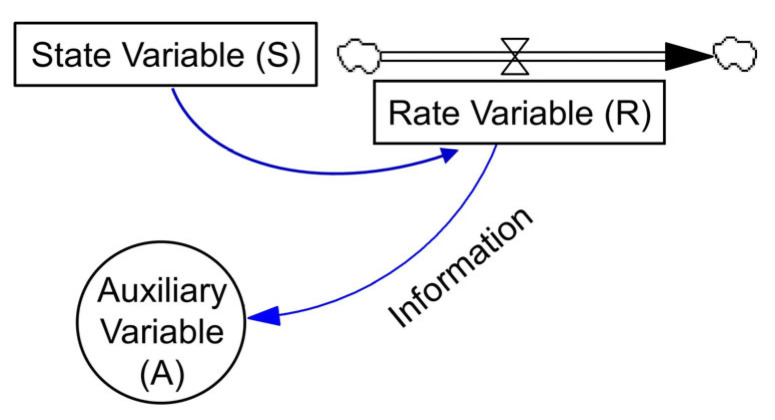
The basic structure elements of SD model.

**Figure 4 entropy-25-00181-f004:**
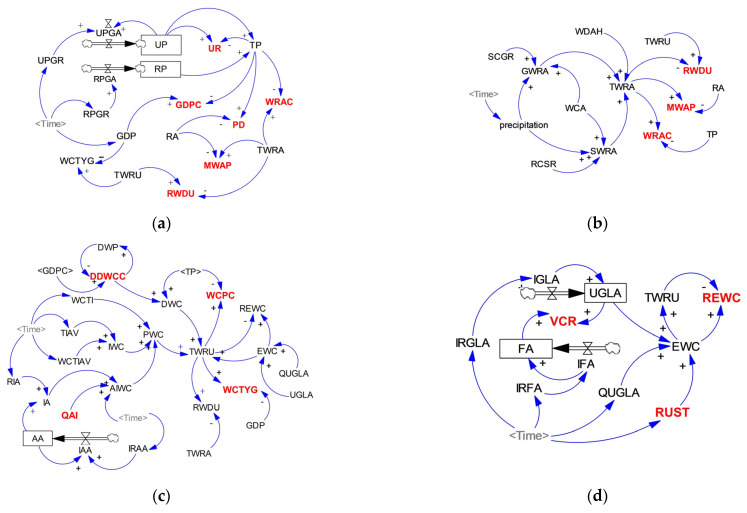
Relationship structure and feedback features of WSEE system by SD model: (**a**) social economy subsystem; (**b**) water resources supply subsystem; (**c**) water resources utilization subsystem; (**d**) ecological environment subsystem. Note: the variables presented in bold and red color are utilized as the evaluation indicators of ODCNE model of WSEE system in this study.

**Figure 5 entropy-25-00181-f005:**
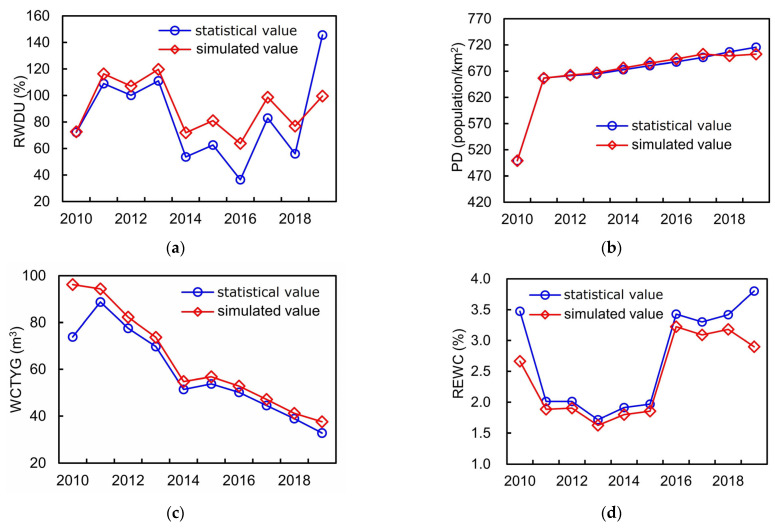
Simulated and observed values of four typical indicators of WSEE system. (**a**) RWDU; (**b**) PD; (**c**) WCTYG; (**d**) REWC.

**Figure 6 entropy-25-00181-f006:**
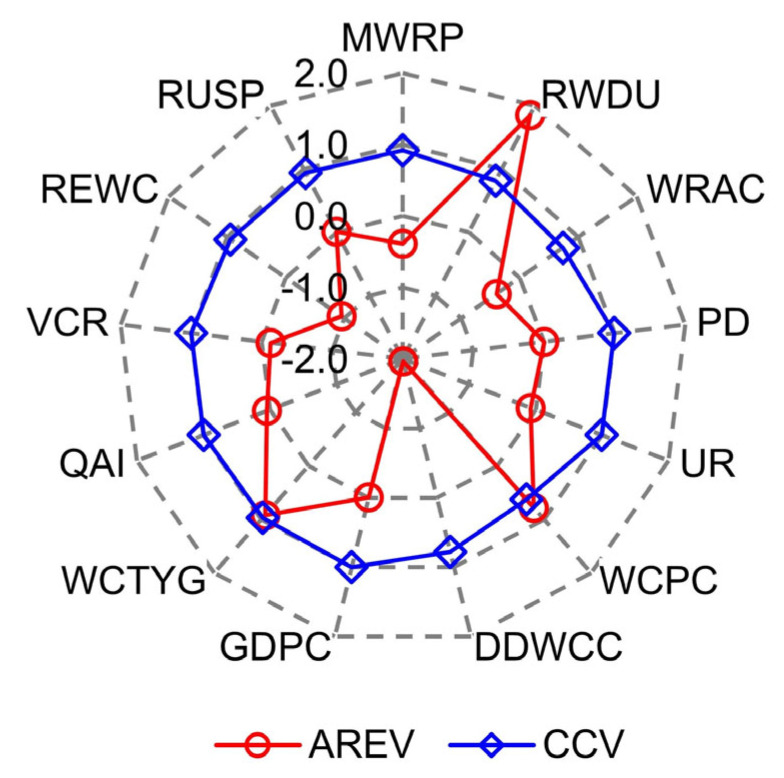
Variation of AREV and CCV of different indicators of WSEE system.

**Figure 7 entropy-25-00181-f007:**
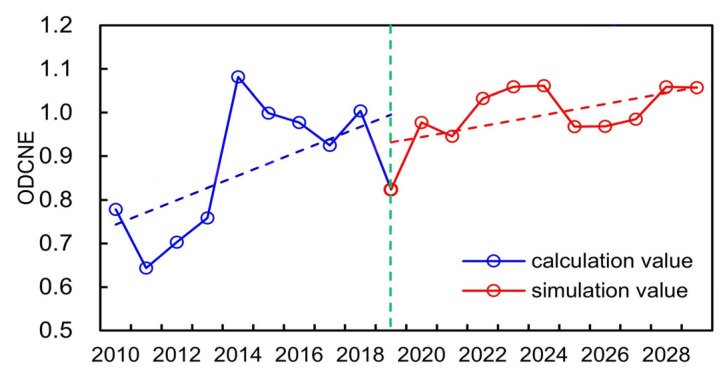
Variation of ODCNE value from 2010 to 2029 in Hefei city.

**Figure 8 entropy-25-00181-f008:**
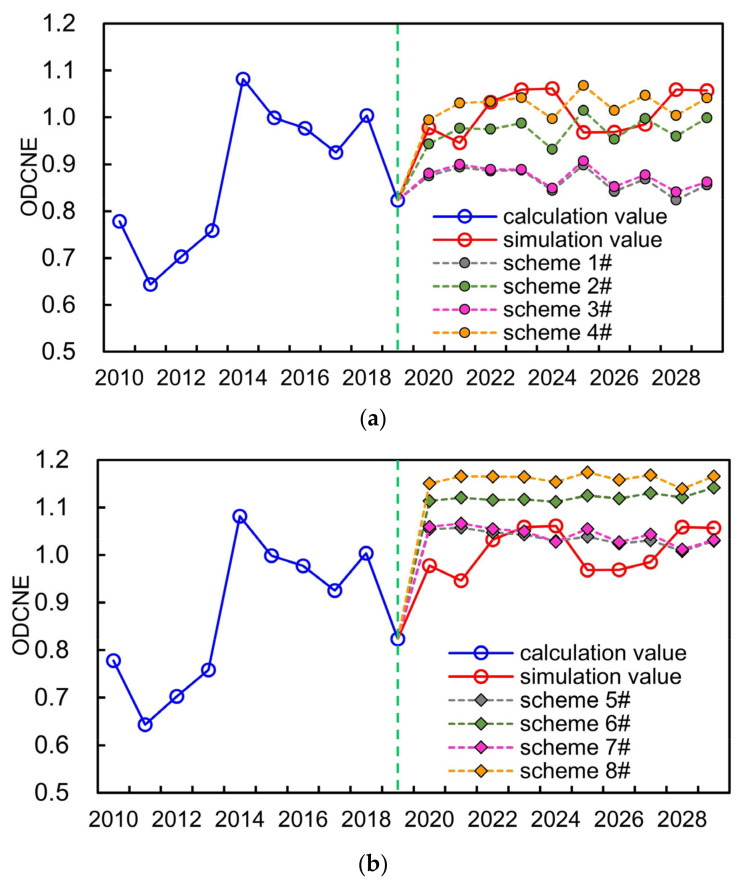
Variation of ODCNE under different development scenarios in Hefei city: (**a**) dry year; (**b**) wet year; (**c**) normal year.

**Figure 9 entropy-25-00181-f009:**
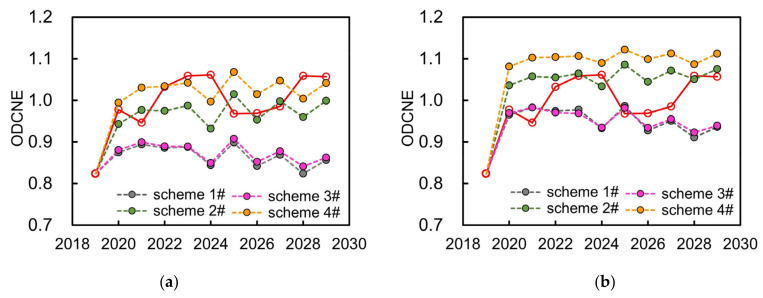
Variation of ODCNE of dry year scenarios before and after the YHD project: (**a**) before YHD project; (**b**) after YHD project.

**Table 1 entropy-25-00181-t001:** Evaluation indicators and grade dividing standard of WSEE system in Hefei city.

Subsystem	Indicator	Abbreviation	Unit	Threshold of Carry Criteria			Attribute	Weight
				Grade I (Loadable)	Grade II (Critical)	Grade III (Unloadable)		
Water resources subsystem	***S***_1_: modulus of water resources production	MWRP	10^4^ m^3^/km^2^	≥35	35~20	<20	positive	0.1446
	***S***_2_: rate of water resources development and utilization	RWDU	%	≤70	70~100	>100	negative	0.0631
	***S***_3_: water resources availability per capita	WRAC	m^3^	≥580	580~440	<440	positive	0.1446
Society subsystem	***S***_4_: population density	PD	population/km^2^	≤550	550~700	>700	negative	0.0860
	***S***_5_: urbanization rate	UR	%	≤69	69~78	>78	negative	0.0686
	***S***_6_: water consumption per capital	WCPC	m^3^/year	≤310	310~380	>380	negative	0.1144
	***S***_7_: daily domestic water consumption per capital	DDWCC	L	≤180	180~230	>230	negative	0.0430
Economy subsystem	***S***_8_: GDP per capita	GDPC	10^3^ yuan	≥150	150~95	<95	positive	0.0525
	***S***_9_: water consumption of ten thousand Yuan GDP	WCTYG	m^3^	≤40	40~70	>70	negative	0.0859
	***S***_10_: quota of agricultural irrigation	QAI	m^3^/mu	≤260	260~330	>330	negative	0.0859
Ecological environment subsystem	***S***_11_: vegetation coverage rate	VCR	%	≥20	20~10	<10	positive	0.0305
	***S***_12_: rate of ecological water consumption	REWC	%	≥10	10~5	<5	positive	0.0314
	***S***_13_: rate of urban sewage treatment	RUST	%	≥90	90~70	<70	positive	0.0492

**Table 2 entropy-25-00181-t002:** Estimated values and equations of parameters under different future scenarios of WSEE system.

Scenarios	Parameters	Wet Year	Normal Year		Dry Year
water resources supply	precipitation	random within [1160, 1600]	random within [971, 1160]		random within [800, 971]
water resources utilization		**water saving implement (high)**		**water saving implement (low)**	
	DDWCC	((GDPC/365) × 0.001502 + 0.352431)/DWP × 1000		((GDPC/365) × 0.001702 + 0.352431)/DWP × 1000	
	WCTI	0.5408 × ln(Year − 2009) + 0.8485		0.1645 × (Year − 2009) + 0.9538	
	QAI	245		260	
	WCTIAV	15.6		20	
social economy development		**economy development (low)**		**economy development (high)**	
	UPGR	random within [0, 0.0154]		random within [0.01, 0.03]	
	TIAV	(167 × (Year − 2009) + 56.8)/10000		(180 × (Year − 2009) + 56.8)/10000	
	GDP	2083 × ln(Year − 2009) + 2301		650 × (Year − 2009) + 2371	
	industrial water utilization price	random within [0.1, 0.2]		random within [0.15, 0.25]	

**Table 3 entropy-25-00181-t003:** Establishment of simulation schemes under different scenarios in Hefei city.

Number	Water Resources Supply			Social Economy Development		Water Resources Utilization	
	Wet	Normal	Dry	High	Low	High	Low
scheme 1#			★		★		★
scheme 2#			★		★	★	
scheme 3#			★	★			★
scheme 4#			★	★		★	
scheme 5#	★				★		★
scheme 6#	★				★	★	
scheme 7#	★			★			★
scheme 8#	★			★		★	
scheme 9#		★			★		★
scheme 10#		★			★	★	
scheme 11#		★		★			★
scheme 12#		★		★		★	

Note: the sign ★ represented the corresponding scenarios of different schemes.

**Table 4 entropy-25-00181-t004:** Average of ODCNE under different development scenarios in the future.

Scenario	Dry Year	Wet Year	Normal Year
low economy development and low water saving	scheme 1#/0.8674	scheme 5#/1.0364	scheme 9#/0.9456
low economy development and high water saving	scheme 2#/0.9736	scheme 6#/1.1223	scheme 10#/1.0503
high economy development and low water saving	scheme 3#/0.8754	scheme 7#/1.0426	scheme 11#/0.9493
high economy development and high water saving	scheme 4#/1.0278	scheme 8#/1.1608	scheme 12#/1.0964
average	0.9361	1.0905	1.0104

## Data Availability

Data will be made available on request.

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
