# Peer review of "Structure Simulation and Equilibrium Evaluation Analysis of Regional Water Resources, Society, Economy and Ecological Environment Complex System"

_entropy, 2023, doi:10.3390/e25020181_

Round 1

Reviewer 1 Report

A very interesting manuscript, dealing with the analysis of the WSEE system in Hefei city and implementing an evaluation system through the ordered degree and connection number entropy coupling model.

The paper is well presented and organized; english writing is good and according to the Reviewer can be accepted for publication. The Reviewer suggests to adopt only one acronym for the water resources, society, economy and ecological environment complex system: the Authors sometimes write WSEE and in other parts of the paper WESE.

Author Response

Responses to the Comments of Reviewer #1:
1.    A very interesting manuscript, dealing with the analysis of the WSEE system in Hefei city and implementing an evaluation system through the ordered degree and connection number entropy coupling model. The paper is well presented and organized. English writing is good and according to the Reviewer can be accepted for publication. The Reviewer suggests to adopt only one acronym for the water resources, society, economy and ecological environment complex system: the Authors sometimes write WSEE and in other parts of the paper WESE.
Response: Thanks very much for the reviewers’ valuable comments, we have repeatedly checked the content of the manuscript, and the acronym of water resources, society, economy and ecological environment complex system has been uniformly modified as WSEE, please refer to the revised manuscript for more details.

Reviewer 2 Report

A very difficult work to read, probably even more so for common readers.
The sentences are extremely long and thought in a different language and way than English. This discourage the reader.
The imputation data cannot be verified anywhere, the authors must be taken at their word...
There are some small discrepancies between the parameters represented (observed and simulated) on some graphs, insufficiently explained.
Some scenarios are a bit risky/futuristic given the current state of the environment and technologies in China and many developing countries of the world. Good thing the scenarios are not exaggerated in the very long term, as in the case of climatologists.
The bibliography is excessively based on the Chinese one, less on other categories, and those not very recent.

Please, see the attached     entropy-2017028-peer-review-v1.pdf     with several specifications.

Author Response

Responses to the Comments of Reviewer #2:
1.    A very difficult work to read, probably even more so for common readers. The sentences are extremely long and thought in a different language and way than English. This discourage the reader.
Response: Thanks very much for the reviewers’ valuable comments. We have repeatedly checked the gramma and sentences of the manuscript, and according to the notes and comments provided by reviewers, many long sentences have been simplified or divided into several short parts. Besides, some writing mistakes and descriptions have been updated, and all the modifications have been colored in red, please refer to the revised manuscript for more details, and hoping that all the revisions will meet the requirements of the reviews.
2.    The imputation data cannot be verified anywhere, the authors must be taken at their word...There are some small discrepancies between the parameters represented (observed and simulated) on some graphs, insufficiently explained.
Response: Thanks very much for the reviewers’ valuable comments. As presented in chapter 4.1, in Figure. 5, the values of RWDU, PD, WCTYG and REWC, as shown by blue curve, are historical observed data from 2010 to 2019, and the corresponding values shown by red curve are simulated value of different typical indicators by means of SD method. The simulation error of different indicators is discussed utilizing two error parameters of AREV and CCV, as indicated in Figure 6. In detail, (1) the simulated and observed data series presented obvious correlation features with the CCV value all exceeding 0.81, and (2) the average of CCV for entire 13 indicators is 0.78, which falls within the acceptable range of correlation verification. Besides, the previous simulation value has been modified as simulated value in Figure 5, please refer to the revised manuscript from line 455 to 472 for more details.
3.    Some scenarios are a bit risky/futuristic given the current state of the environment and technologies in China and many developing countries of the world. Good thing the scenarios are not exaggerated in the very long term, as in the case of climatologists.
Response: Thanks very much for the reviewers’ valuable comments. As indicated in Table. 3 and 4, the development schemes of future scenarios are established according to different combinations of water resources supply, utilization and social economy development subsystems. And the values of related parameters, indicators and equations, as shown in Table 2, are determined based on the parameter estimation results and government planning report. Indeed, some scenarios and schemes seem unrealistic or futuristic given the current development state, further recognition and exploration for the future scenarios will be conducted in the future work.
4.    The bibliography is excessively based on the Chinese one, less on other categories, and those not very recent. Please, see the attached entropy-2017028-peer-review-v1.pdf with several specifications.
Response: Thanks very much for the reviewers’ valuable comments. Four Chinese references, No.2, 21, 32 and 35 (line 673, 712, 735 and 741), have been updated with English references. Besides, according to the notes and comments in the attached file entropy-2017028-peer-review-v1.pdf, many long sentences have been simplified or divided, some concerning issues are all addressed, and all the modified have been colored in red in the revised manuscript, please refer to the revised manuscript for more details.

Reviewer 3 Report

The manuscript deals with the crucial issue of the tructure simulation and equilibrium evaluation analysis of regional water resources, society, economy and ecological environment complex system. Authors applied the information entropy. ordered degree and connection number coupling method to reveal the membership characteristics between different equilibrium indicators and evaluation criterion. Meanwhile, the system dynamics approach was introduced to describe the relationship features among different equilibrium subsystems. And eventually, the ordered degree, connection number, information entropy and system dynamics integrated model was established to conduct relationship structure simulation and evolution trend evaluation of WSEE system. Comments: If possible add references to equations presented in the text. Authors do not explain well, where is the novelty of the distinguished method. Conclusions are not sufficiently described. How in practice the results of the presented work can be used? This should be discussed in the point concerning discussion of the results and presented  in the conclusions. The manuscript should present the value added with respect to existing methods. The last point of the article contains in fact only the conclusions relating to the researched case study, but there is no more detailed perspective. Authors can add future perspective concerning  developing predictive models, as presented in references: A Case Study in View of Developing Predictive Models for Water Supply System Management. Energies. 2021; 14(11):3305. https://doi.org/10.3390/en14113305 and Water Network-Failure Data Assessment. Energies 2020, 13, 2990. https://doi.org/10.3390/en13112990. Besides, there is no discussion about possible limitations of using the proposed modelling. On what base the ranges of parameter values and indicators under different scenarios of WSEE system were assumed? Much more consistency needs to be achieved in the interplay between results and conclusions.

Author Response

Responses to the Comments of Reviewer #3:
The manuscript deals with the crucial issue of the structure simulation and equilibrium evaluation analysis of regional water resources, society, economy and ecological environment complex system. Authors applied the information entropy. ordered degree and connection number coupling method to reveal the membership characteristics between different equilibrium indicators and evaluation criterion. Meanwhile, the system dynamics approach was introduced to describe the relationship features among different equilibrium subsystems. And eventually, the ordered degree, connection number, information entropy and system dynamics integrated model was established to conduct relationship structure simulation and evolution trend evaluation of WSEE system. Comments: 
1.    If possible add references to equations presented in the text. 
Response: Thanks very much for the reviewers’ valuable comments, the related references of equations (1)~(10) have all been provided in the revised manuscript, and the computation equations of different indicators shown in Table 2 were determined through parameter fitting and regression calculation by means of system dynamics method in this manuscript.
2.    Authors do not explain well, where is the novelty of the distinguished method. 
Response: Thanks very much for the reviewers’ valuable comments, the novelty of the manuscript has been provided, i.e., the combination of ordered degree and information entropy methods was also the primary novelty of this study, which was expected to incorporate the overall influence of uncertainties from the perspectives of identity, difference and opposition, please refer to line 98 to 101 of the revised manuscript for more details.
3.    Conclusions are not sufficiently described. How in practice the results of the presented work can be used? This should be discussed in the point concerning discussion of the results and presented in the conclusions. 
Response: Thanks very much for the reviewers’ valuable comments, the practical application of the integrated simulation and evaluation analysis results of WSEE system was provided in the revised manuscript, i.e., …the integrated simulation and evaluation analysis results of WSEE system will provide reasonable decision-making basis for the development of annual water resource allocation schemes of YHD project, please refer to line 647 to 649 for more details. Besides, the practical guidelines and principles for the development of annual water resource allocation schemes of YHD project combining different future development scenario simulation results of WSEE system will be further discussed in future work.
4.    The manuscript should present the value added with respect to existing methods. The last point of the article contains in fact only the conclusions relating to the researched case study, but there is no more detailed perspective. Authors can add future perspective concerning developing predictive models, as presented in references: A Case Study in View of Developing Predictive Models for Water Supply System Management. Energies. 2021; 14(11):3305. https://doi.org/10.3390/en14113305 and Water Network-Failure Data Assessment. Energies 2020, 13, 2990. https://doi.org/10.3390/en13112990. 
Response: Thanks very much for the reviewers’ valuable comments. The primary novelty and motivation of the manuscript with respect to existing methods have been provided in the revised manuscript, i.e., …the combination of ordered degree and information entropy methods was also the primary novelty of this study, …but neglected to reveal the mutual feedback relationship and structure features between different elements of WSEE system, these are all fundamental tasks to simulate the future evolution trend of WSEE system, and also the motivation of this study. More details can be referred from line 68 to 70, and 98 to 101 in the revised manuscript. Besides, more work focusing on the establishment and recognition of future development scenarios combining the abovementioned references will be conducted in the future work, and the first reference, A Case Study in View of Developing Predictive Models for Water Supply System Management. Energies. 2021; 14(11):3305. https://doi.org/10.3390/en14113305, has been cited in the revised manuscript, please refer to line 656 for more details.
5.    Besides, there is no discussion about possible limitations of using the proposed modelling. On what base the ranges of parameter values and indicators under different scenarios of WSEE system were assumed? Much more consistency needs to be achieved in the interplay between results and conclusions.
Response: Thanks very much for the reviewers’ valuable comments, the future research direction and limitation of the proposed framework has been provided in the revised manuscript, i.e., how to construct reasonable future development scenarios and then conduct effective and quantitative expression of the nonlinear feedback relationships between different equilibrium subsystem and indicators, are also limitations and big challenge, which is also the future research direction of this study, please refer to line 653 to 657 of the revised manuscript for more details. Besides, as for Table. 2, the values of different parameters and equations of different future scenarios are determined based on the historical observed data of WSEE system as well as government planning report, 2020 in Anhui province, which has been stated in the revised manuscript, please refer to line 524 to 526 for more details.

Round 2

Reviewer 2 Report

In the structure of the text, there are still a few omissions (e.g. Yuans instead of Dollars, social ... and other different types of economy, some science-fiction scenarios etc.), but overall, the work has been improved/adjusted.
Also, the bibliography is still overwhelmingly dominated by Chinese literature. Does this mean that others don't matter ...? Perhaps, it would not be that bad for the authors to add a few essential non-Chinese bibliographic titles on the addressed topics.

Author Response

Thanks very much for the reviewers' vluable comments. (1) We have carefully modified the manuscript and unified some expressions, such as social, society, economy, economic etc. (2) In terms of the bibliography, we further updated the previous Chinese references with English articles, and totally 10 Chinese references (No. 4, 7, 9, 11, 13, 15, 19, 33, 36 and 41) were replaced. All the modifications have been clorred in red in the revised manuscript.

Reviewer 3 Report

The authors took into account the comments of the reviewers. I think the work is worth publishing

Author Response

Thanks very much for the reviewers' valuable comments and suggestions.